# Potential Anti-SARS-CoV-2 Activity of Pentosan Polysulfate and Mucopolysaccharide Polysulfate

**DOI:** 10.3390/ph15020258

**Published:** 2022-02-21

**Authors:** Fuming Zhang, Peng He, Andre L. Rodrigues, Walter Jeske, Ritesh Tandon, John T. Bates, Michael A. Bierdeman, Jawed Fareed, Jonathan Dordick, Robert J. Linhardt

**Affiliations:** 1Department of Chemical and Biological Engineering, Center for Biotechnology and Interdisciplinary Studies, Rensselaer Polytechnic Institute, Troy, NY 12180, USA; lopesa@rpi.edu (A.L.R.); dordick@rpi.edu (J.D.); 2Departments of Chemistry and Chemical Biology, Center for Biotechnology and Interdisciplinary Studies, Rensselaer Polytechnic Institute, Troy, NY 12180, USA; hep3@rpi.edu; 3Department of Pathology and Laboratory Medicine, Loyola University Medical Center, Maywood, IL 60153, USA; wjeske@luc.edu (W.J.); jfareed@luc.edu (J.F.); 4Department of Microbiology and Immunology, University of Mississippi Medical Center, Jackson, MS 39216, USA; rtandon@umc.edu (R.T.); jtbates@umc.edu (J.T.B.); 5Department of Medicine, University of Mississippi Medical Center, Jackson, MS 39216, USA; mbierdeman@umc.edu; 6Departments of Biological Science and Biomedical Engineering, Center for Biotechnology and Interdisciplinary Studies, Rensselaer Polytechnic Institute, Troy, NY 12180, USA

**Keywords:** SARS-CoV-2, pentosan polysulfate, mucopolysaccharide polysulfate, heparan sulfate, heparin, surface plasmon resonance

## Abstract

With the increased prevalence of new SARS-CoV-2 variants of concern, such as Delta and Omicron, the COVID-19 pandemic has become an ongoing human health disaster, killing millions worldwide. SARS-CoV-2 invades its host through the interaction of its spike (S) protein with a host cell receptor, angiotensin-converting enzyme 2 (ACE2). In addition, heparan sulfate (HS) on the surface of host cells plays an important role as a co-receptor for this viral pathogen–host cell interaction. Our previous studies demonstrated that many sulfated glycans, such as heparin, fucoidans, and rhamnan sulfate have anti-SARS-CoV-2 activities. In the current study, a small library of sulfated glycans and highly negatively charged compounds, including pentosan polysulfate (PPS), mucopolysaccharide polysulfate (MPS), sulfated lactobionic acid, sulodexide, and defibrotide, was assembled and evaluated for binding to the S-proteins and inhibition of viral infectivity in vitro. These compounds inhibited the interaction of the S-protein receptor-binding domain (RBD) (wild type and different variants) with immobilized heparin, a highly sulfated HS, as determined using surface plasmon resonance (SPR). PPS and MPS showed the strongest inhibition of interaction of heparin and S-protein RBD. The competitive binding studies showed that the IC_50_ of PPS and MPS against the S-protein RBD binding to immobilized heparin was ~35 nM and ~9 nM, respectively, much lower than the IC_50_ for soluble heparin (IC_50_ = 56 nM). Both PPS and MPS showed stronger inhibition than heparin on the S-protein RBD or spike pseudotyped lentiviral particles binding to immobilized heparin. Finally, in an in vitro cell-based assay, PPS and MPS exhibited strong antiviral activities against pseudotyped viral particles of SARS-CoV-2 containing wild-type or Delta S-proteins.

## 1. Introduction

The ongoing COVID-19 pandemic, caused by acute respiratory syndrome-related coronavirus 2 (SARS-CoV-2), has resulted in a public health disaster and led to millions of deaths globally. Many variants of SARS-CoV-2 have been recognized by the World Health Organization (Alpha, Beta, Gamma, Delta, Omicron, etc.) since the beginning of the pandemic. These variants of concern (VOC) have exhibited increased transmissibility, virulence, and/or a reduced effectiveness of vaccines, resulting in immune breakthrough infections [1,2,3]. Up until November 2021, the Delta variant (B.1.617.2) had been the most common COVID-19 variant circulating worldwide since October 2020 [4,5]. With greatly increased transmissibility, however, a new variant, Omicron (B.1.1.529), has now become the predominant strain [5,6]. Although there are several vaccines offering protection for COVID-19, the efficacy of these vaccines for such VOC has been reduced [7]. Supplementing vaccines, some oral or injectable therapeutics have been developed (or repurposed), including remdesivir, favipiravir, simeprevir, various monoclonal antibodies, and most recently paxlovid and molnupiravir, which can inhibit the infection or propagation of SARS-CoV-2 [8,9]. However, the clinical efficacies of many of these agents are highly variable and are most effective only in the first few days of infection [10,11,12,13]. Moreover, apart from monoclonal antibodies, their prophylactic use is not indicated. Therefore, new effective drugs for both therapeutic and critically prophylactic uses to combat COVID-19 are desperately needed. 

Glycosaminoglycans (GAGs) are a family of highly negatively charged linear polysaccharides including heparin/heparan sulfate (HS), chondroitin sulfate (CS)/dermatan sulfate (DS), hyaluronan (HA), and keratan sulfate (KS) [14]. GAGs interact with various proteins, such as growth factors/receptors, morphogens, chemokines, extracellular matrix proteins, lipoproteins, and pathogens. These interactions play vital roles in pathological processes/diseases such as inflammation, angiogenesis, cancer, neurodegenerative diseases, and infectious diseases [15,16]. GAG–protein interactions have been targeted for many therapeutic applications [17,18]. During the initial stage of host cell invasion, SARS-CoV-2 invades the human host cells through the interaction of its spike (S) protein with a host cell receptor, angiotensin-converting enzyme 2 (ACE2) [16]. In addition, heparan sulfate (HS) on the surface of host cells plays an important role as a co-receptor for this viral pathogen–host cell interaction [19,20,21,22,23]. It is known that HS functions as a crucial cofactor for SARS-CoV-2, binding to ACE2 by interacting with the receptor-binding domain (RBD) at the S1 subunit of the S-protein, which facilitates the opening of S-protein conformation for ACE2 binding [22].

We and others have shown that sulfated polysaccharides inhibit the interaction between HS and the S-protein, including heparin, fucoidans, and rhamnan sulfate [24,25,26,27,28]. In the current study, a small library of sulfated glycans and highly negative compounds (see Table 1 and Figure 1), including pentosan polysulfate (PPS), mucopolysaccharide polysulfate (MPS), sulfated lactobionic acid, sulodexide, and defibrotide, was assembled and evaluated for binding to the WT and variant S-proteins and inhibition of viral infectivity in vitro. PPS is a heparin mimetic with a highly sulfated polysaccharide backbone [29]. It is synthesized through the chemical sulfonation of a plant-derived β-(1 → 4)-xylan. PPS is an FDA-approved active pharmaceutical ingredient of the oral drug Elmiron™. MPS is a semisynthetic glycosaminoglycan with a backbone that is isolated from mammalian cartilage before its chemical sulfation [30]. Surface plasmon resonance (SPR) was employed for the direct, label-free, real-time quantification of the binding of these sulfated glycans to the SARS-CoV-2 S-protein. Furthermore, *in vitro* data on pseudotyped viral particle infection of cultured cells was used to compare PPS and MPS binding affinity to S-protein to that of heparin (as positive control) as potential therapeutic or prophylactic agents to combat COVID-19.

## 2. Results and Discussion

### 2.1. Interaction between SARS-CoV-2 S-Protein RBD and Heparin

We demonstrated previously [22] that heparin interacts with the SARS-CoV-2 S-protein with high avidity and proposed that HS facilitates host cell entry of SARS-CoV-2 as a co-receptor of ACE2, which has been confirmed by other researchers [23,24]. Protein binding to heparin/HS results from the ionic- or hydrogen-bonding interactions of basic amino acid residues, placed in defined motifs [31,32,33,34] within the protein, with the anionic carboxyl and sulfo groups within these GAGs [33]. With the greater prevalence of new SARS-CoV-2 VOC, additional S-protein mutants were identified [35]. As the primary antigen of SARS-CoV-2, mutations of the S-protein greatly alter the viral infectivity, disease severity, and effectiveness of vaccines [35]. In the current study, SPR was applied to measure the binding kinetics and affinity of SARS-CoV-2 S-protein RBD (WT and different VOC) interaction with heparin using a sensor chip with immobilized heparin. Sensorgrams of S-protein RBD (WT and VOC) interactions with heparin are shown in Figure 2. The resulting sensorgrams were used to determine binding kinetics and affinity (i.e., association rate constant: ka; dissociation rate constant: kd; and binding equilibrium dissociation constant: K_D_, K_D_ = kd/ka) by globally fitting the entire association and dissociation phases using the 1:1 Langmuir binding model from T200 Evaluation software. Binding kinetic parameters (ka and kd) and affinity (K_D_) were calculated (Table 2) from sensorgrams and globally fitted to the 1:1 Langmuir model from T200 Evaluation software. The binding kinetics and affinities of the different VOC S-protein RBDs were comparable to the WT version except for N501Y, which showed higher affinity to heparin, and L452R, which showed lower affinity to heparin.

### 2.2. SPR Solution Competition Study on the Inhibition of Sulfated Glycans to the Interaction between Surface-Immobilized Heparin with S-Protein

Solution/surface competition experiments were performed using SPR to examine the inhibition of sulfated glycans to the interaction between heparin (on surface) with S-protein. PPS and MPS potently inhibited the S-protein–heparin interaction, while sulfated lactobionic acid, GL-288-Y-1, GL-522-Y-1, and GL-522Y-1 calcium exhibited a modest inhibition activity (Figure 3). Relatively lower inhibition activity of different versions of defibrotide was observed. Based on these data, PPS and MPS were selected for further investigation. Importantly, the FDA has approved PPS as an oral anti-thrombotic agent for the management of patients with interstitial cystitis, and it is also used for clinical disorders such as antagonism of enzymatic activities and inhibition of HIV infectivity [29,36,37]. MPS has been used for the topical treatment of superficial phlebitis, hematomas, and sports-related injuries [30].

### 2.3. IC_50_ Measurement of the Inhibition of S-Protein Binding to Heparin by PPS and MPS Using Solution Competition SPR

Solution competition dose–response analysis between surface-immobilized heparin and soluble PPS and MPS was performed to calculate IC_50_ values and to quantify the inhibition by PPS and MPS of the interaction between heparin (on surface) and S-protein RBD (WT) (Figure 4). S-protein RBD was pre-mixed with different concentrations of PPS, MPS, or heparin before injection into the heparin chip. When the active binding sites on the S-protein RBD were occupied by glycan in solution, its binding to the surface-immobilized heparin decreased, resulting in a reduction in signal in a concentration-dependent fashion (Figure 4). The IC_50_ values were calculated from the plots S-protein RBD binding signal (normalized) versus glycan concentration in solution. The competitive binding studies revealed that the IC_50_ of PPS and MPS against the S-protein RBD binding to immobilized heparin was ~35 nM and ~9 nM, respectively, which was much lower than the IC_50_ for heparin (IC_50_ = 56 nM). This could be due to the level of sulfation being higher for MPS and PPS compared with heparin. The average heparin disaccharide contains ~2.7 sulfo groups, while MPS disaccharide has >4 sulfo groups and PPS disaccharide has >3 sulfo groups. Based on a recent study [38], the 2,3-disulfated polyxylan oligosaccharide is the key sugar moiety of PPS’ binding to the S-protein RBD. MPS has a heparin-like molecular structure with a high level of sulfo groups, allowing this glycan to interact with the S-protein.

### 2.4. Inhibition of S-Protein RBD Mutants’ Binding to Heparin by PPS and MPS Using Solution Competition SPR

Solution/surface competition experiments were also performed using SPR to examine the inhibition of different S-protein RBD VOC–heparin interactions by PPS and MPS (Figure 5). Using the same concentration of PPS, MPS, and heparin (5 ng/mL), PPS and MPS showed stronger inhibition of most of S-protein RBD mutants tested than heparin, with the exception of the L452R + T478K mutant, which showed comparable inhibition to heparin.

### 2.5. SPR Solution Competition Study on the Inhibition of the Interaction between Heparin and Pseudotyped Viral Particles by PPS and MPS

Solution/surface competition experiments were carried out to test the inhibition of pseudovirus particle (wild-type and Delta variant)–heparin interaction by PPS and MPS (Figure 6). Both PPS and MPS showed stronger inhibition of both wild-type and Delta variant pseudotyped viral particle binding to heparin surface compared with heparin control.

### 2.6. In Vitro SARS-CoV-2 Pseudotyped Viral Particle Neutralization

Finally, an in vitro SARS-CoV-2 pseudotyped viral particle neutralization assay was performed. Sulfated glycans such as heparin have been shown to inhibit viral infection by interacting with the SARS-CoV-2 spike protein [24,25,26,27]. Thus, we investigated the ability of PPS and MPS to inhibit viral particle entry using a cell-based neutralization assay. HEK293T cells were used that stably expressed the ACE2 receptor (HEK293T-ACE2). Six different concentrations were tested at 1:10 viral dilution, thus enabling the determination of an IC_50_ value for viral inhibition based on the expression of EGFP as a marker for functional viral entry. Briefly, PPS or MPS was incubated with pseudovirus particles for 1 h at 37 °C, after which the mixture was added to the HEK293T-ACE2 cells and incubated for 4 h. These incubation steps were performed under serum-free conditions, as sera often contain growth factors that interact with polysaccharides, thus interfering with its interaction with the spike protein. After the 4 h incubation, there was a medium exchange with serum to sustain cell growth for 48 h, after which the plates were assayed for expression of EGFP. The results of the neutralization experiment are shown in Figure 7. For each concentration, the percentage of infected cells was normalized to the percentage of infected cells relative to the control (no PPS and MPS and 1:10 viral dilution). The lowest dilution (PPS and MPS concentration of 1 μg/mL) provided > 80% inhibition of viral particle entry for both the WT and Delta variant. The IC_50_ values of PPS for the WT and Delta variant were 0.45 and 0.07 µg/mL, and the IC_50_ values of MPS for the WT and Delta variant were 0.42 and 0.28 µg/mL, respectively.

## 3. Materials and Methods

### 3.1. Materials

Fourteen sulfated glycans (Table 1) were collected from their manufacturers in Dr. Fareed’s Lab. SARS-CoV-2 S-protein RBD wild type (WT) and N501Y were expressed in Expi293F cells provided by the Bates Lab, University of Mississippi Medical Center. SARS-CoV-2 S-protein RBD mutants (related to Delta variants of SARS-CoV-2) were purchased from Sino Biological US Inc. (Wayne, PA, USA). SARS-CoV-2 pseudoviral particles (WT and Delta variant) were prepared in Tandon’s Lab as previously described [27]. Sensor SA chips were from Cytiva (Uppsala, Sweden). SPR measurements were performed on a BIAcore 3000 operated using BIAcore 3000 or T200 SPR (Cytiva, Uppsala, Sweden). The cell line HEK293T was purchased from American Type Culture Collection (ATCC, Manassas, VA, USA).

### 3.2. SPR Measurements of Interactions between Heparin and S-Proteins

Preparation of heparin SPR chip [31]: Heparin (2 mg) and amine-PEG3-Biotin (2 mg, Pierce, Rockford, IL, USA) were dissolved in 200 µL H_2_O; 10 mg NaCNBH_3_ was added. The reaction mixture was heated at 70 ℃ for another 24 h; after that, a further 10 mg NaCNBH_3_ was added, and the reaction was heated at 70 ℃ for another 24 h. The mixture was desalted using the spin column (3000 MWCO). Biotinylated heparin was freeze-dried for heparin chip preparation. In brief, a 20 µL solution of biotinylated heparin (0.1 mg/mL) in HBS-EP+ buffer (0.01 M 4-(2-hydroxyethyl)-1-piperazineethanesulfonic acid, 0.15 M NaCl, 3 mM ethylenediaminetetraacetic acid, 0.05% surfactant P20, pH 7.4) was injected over flow cell 2 (FC2) of the SA chip at a flow rate of 10 µL/min. The successful immobilization of heparin was confirmed by the observation of a ~200 resonance unit (RU) increase in the sensor chip. The control flow cell (FC1) was prepared by 1 min injection with saturated biotin.

Binding kinetics and affinity measurement: Different dilutions of S-protein RBD samples in HBS-EP+ buffer were injected at a flow rate of 30 µL/min. At the end of the sample injection, the same buffer was allowed to flow over the sensor surface to facilitate dissociation. After a 3 min dissociation time, the sensor surface was regenerated by injecting with 30 µL of 2 M NaCl. The response was monitored as a function of time (sensorgram) at 25 °C.

### 3.3. Evaluation of the Inhibition Activity of Sulfated Glycans on Heparin–S-Protein RBD Using Solution Competition SPR

Competition studies between surface heparin and different soluble sulfated glycans and highly negative compounds were performed using SPR. In brief, S-protein RBD (250 nM) samples mixed with different concentrations of sulfated glycans in HBS-EP+ buffer were injected over the heparin chip at a flow rate of 30 µL/min, respectively. After each run, dissociation and regeneration were performed as described above. For each set of competition experiments on SPR, a control experiment (only protein) was performed to ensure that the surface was completely regenerated and that the results obtained between runs were comparable. Once the active binding sites on S-protein molecules were occupied by sulfated glycan in the solution, the binding of S-protein to the surface-immobilized heparin was decreased, resulting in a reduction in signal. The same protein samples were also mixed with heparin in HBS-EP+ buffer and were tested to serve as a positive control.

S-protein RBD (250 nM) samples premixed with different concentrations of sulfated glycan (in ½ serial dilutions with HBS-EP+ buffer) were injected over the heparin chip to measure IC_50_ [31]. The IC_50_ values (concentration of competing analyte resulting in a 50% decrease in response units (RU)) were calculated from the plots (S-protein binding signal (normalized) versus sulfated glycan concentration in solution).

### 3.4. SPR Solution Competition Study of the Inhibition Sulfated Glycans on the Interaction of Heparin and SARS-CoV-2 Pseudoviral Particles

SARS-CoV-2 pseudotyped viral particles premixed with sulfated glycans in HBS-EP+ buffer were injected over the heparin chip at a flow rate of 30 µL/min. Similarly, when the active binding sites on the pseudotyped viral particles were occupied by sulfated glycans in solution, the binding of the viral particles to the surface-immobilized heparin decreased, resulting in a reduction in signal in RU.

### 3.5. In Vitro SARS-CoV-2 Pseudotyped Virus Neutralization Assay

ACE2 stable cell line generation: Lentiviral particles containing the ACE2-Puro construct were produced by transfecting 12.3 μg psPAX2 (Addgene # 12260), 2.5 μg pMD2g (#12259), and 14.7 μg pLenti-hACE2-Puro into HEK293T cells using Lipofectamine 2000 according to the manufacturer’s instructions. The plasmids, psPAX2 and pMD2g, were a gift from Didier Trono (École Polytechnique Fédérale de Lausanne, Switzerland). A medium exchange was carried out 24 h after transfection, and 5 mM sodium butyrate (Millipore Sigma, Burlington, MA, USA) was added to the cells in fresh medium. The supernatants from HEK293T cells carrying the lentiviral particles were harvested at 48 h and 72 h. The supernatants were pooled and concentrated using Lenti-X-Concentrator (Takara Bio, Shiga, Japan) according to the manufacturer’s instructions. The concentrated lentiviral particles carrying ACE2-Puro were delivered to HEK293T cells in 6-well tissue culture-treated plates. After 48 h, 4 μg/mL of puromycin was added to Dulbecco’s Modified Eagle Medium (DMEM) + 10% fetal bovine serum (FBS), and a medium exchange was carried out. The cells were passaged to a T-25 flask and maintained in selection pressure (4 μg/mL puromycin) to remove cells lacking the ACE2-Puro construct.

### 3.6. Production of Spike Pseudotyped Viral Particles

HEK293T cells were seeded in two T175 flasks and cultured in DMEM + 10% FBS. At 70–80% confluence, the cells were transfected using Lipofectamine 2000. For production of WT and Delta spike pseudotyped particles, the cells were transfected with 26 μg of psPAX2, 26 μg of pLV-enhanced green fluorescent protein (EGFP) (a gift from Pantelis Tsoulfas (University of Miami, Florida), Addgene plasmid # 36083), and 8.7 μg of pHDM-SARS-CoV2-S (BEI Resources #NR52514) per flask. A medium exchange was performed 24 h after transfection with the addition of 5 mM sodium butyrate (Millipore Sigma, Burlington, MA). The harvest supernatant was collected at 48 h and 72 h and concentrated using Lenti-X-Concentrator according to the manufacturer’s instructions. The resuspended viral samples were stored at −80 °C until use.

SARS-CoV-2 Pseudotyped Virus Neutralization Assay: Six different concentrations of sulfated glycans were prepared at a 10-fold serial dilution from 1000 to 0.01 μg/mL in DMEM + 1% PenStrep and no FBS. Viral samples were then added at 1:10 dilution, and the mixtures were incubated at 37 °C for 1 h. The samples were then added to HEK293T-ACE2 cells, plated in 96-well plates at 15,000 cells/well, and incubated for 4 h at 37 °C. Afterwards, a media change was performed with DMEM + 1% PenStrep + 10% FBS. The cells were cultured for an additional 48 h and were then stained with 5 μg/mL of Hoechst 33342 and imaged using Cellomics Arrayscan XTI. The infection efficiency was then calculated using the Target Activation Bioapplication. The results of the experiment represent the percentage (%) of maximum infectivity that could be obtained for the experiment. This was carried out by normalizing the percentage (%) infected value for each sample by the percentage (%) infected value at the 1:10 dilution and 0 μg/mL of compound.

## 4. Conclusions

We explored the anti-SARS-CoV-2 activity of sulfated glycans with different structures. SPR analysis confirmed the interactions of heparin with the S-protein RBD from wild-type SARS-CoV-2 and several VOC. In a competition SPR assay, PPS and MPS in solution showed remarkable inhibition activity against chip-surface heparin binding with the wild-type S-protein RBD (with a measured ~35 nM and ~9 nM, respectively), which was much lower than the IC_50_ for heparin (IC_50_ = 56 nM). Both PPS and MPS displayed a higher capacity to bind S-protein RBD from VOC, and the pseudotyped viral particles of wild-type/Delta variant, compared with that of heparin. Finally, we confirmed the neutralizing effect of PPS and MPS on SARS-CoV-2 pseudotyped virus in vitro; The IC_50_ values for PPS inhibition of heparin binding to pseudotyped virus of WT and Delta variant were 0.45 and 0.07 µg/mL, respectively, and the IC_50_ values of for MPS were 0.42 and 0.28 µg/mL, respectively. These results suggest the potential use of PPS and MPS as therapeutic and/or preventative antiviral drugs. Future studies are needed to investigate the structure–activity relationships (SAR), bioavailability, and antiviral activity of low molecular weight PPS and MPS.

## Figures and Tables

**Figure 1 pharmaceuticals-15-00258-f001:**
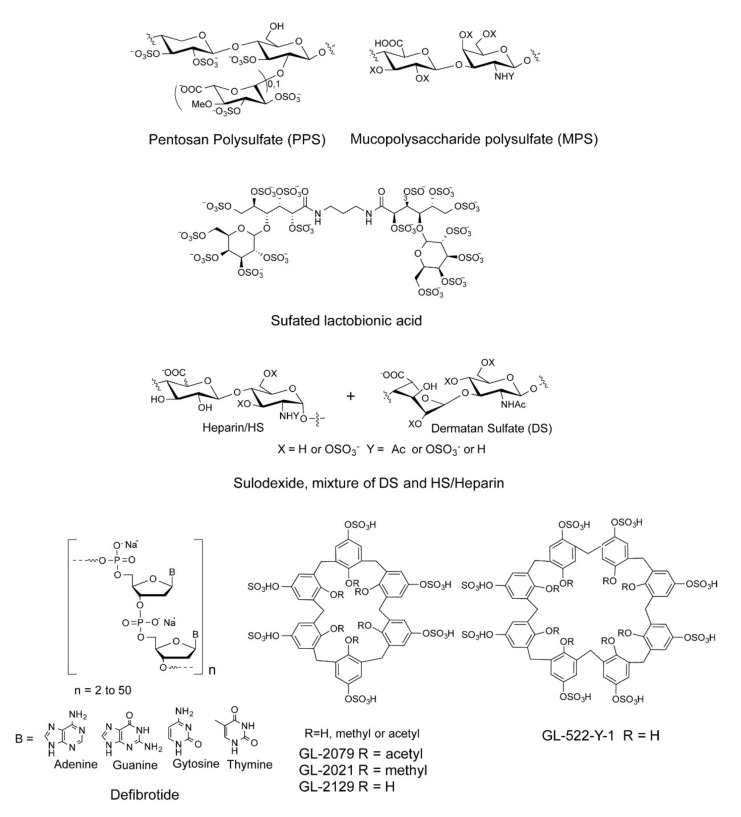
Chemical structures of sulfated glycans and defibrotide.

**Figure 2 pharmaceuticals-15-00258-f002:**
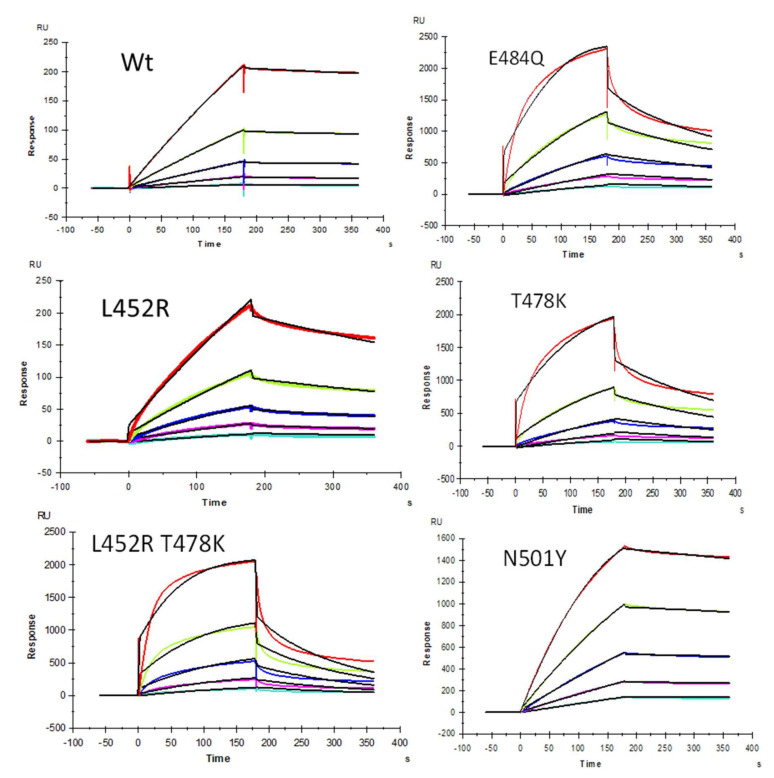
SPR sensorgrams of S-protein RBD wild type (WT) and mutants’ interactions with heparin. Concentration of S-protein mutants (from top to bottom): 1000, 500, 250, 125, and 63 nM, respectively. The black curves are the fitting curves using models from T200 Evaluate software.

**Figure 3 pharmaceuticals-15-00258-f003:**
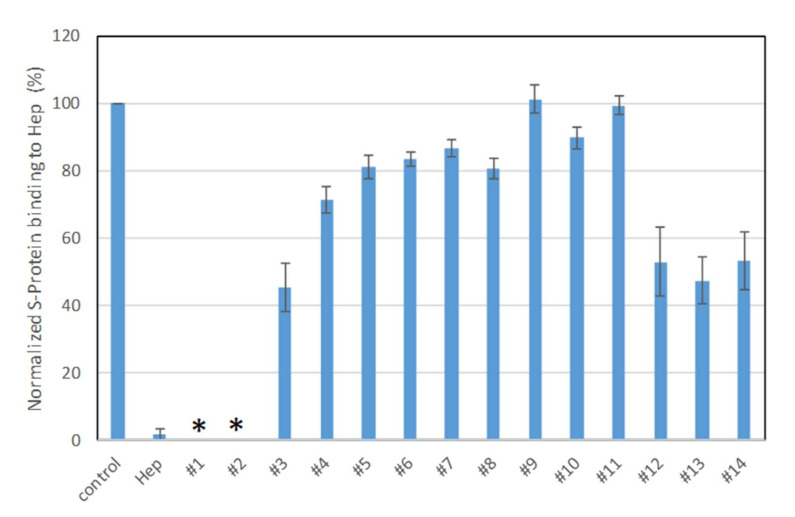
Bar graphs (based on triplicate experiments with standard deviation) of normalized S-protein binding to surface heparin by inhibition with sulfated glycans in solution. Compound #1: PPS, #2: MPS, #3: Sulfated lactobionic acid, #4: Sulodexide, #5–8: Defibrotides, #9–11: 4-t-butylcalix [6] arene-p-sulfonic acids, #12–14: 4-t-butylcalix [8] arene-p-sulfonic acids. * No binding signal detected due to full inhibition by PPS and MPS.

**Figure 4 pharmaceuticals-15-00258-f004:**
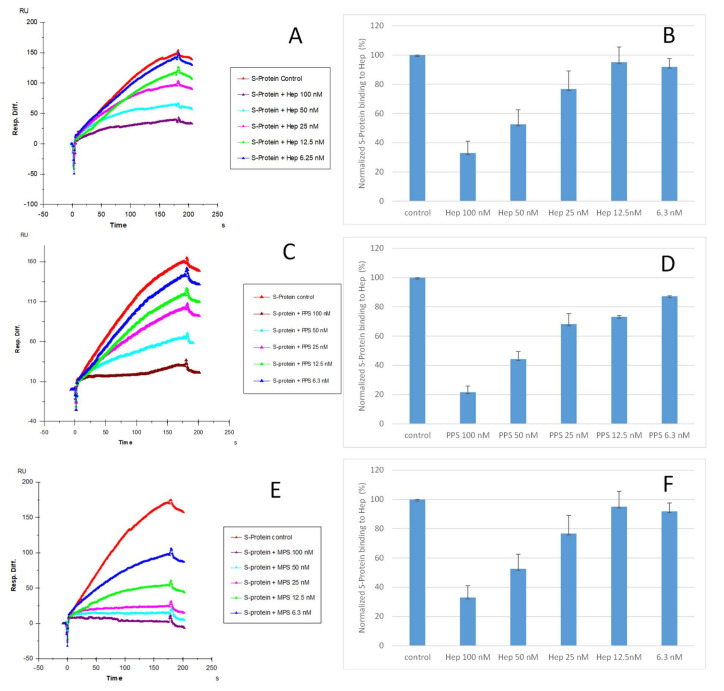
IC_50_ measurement of the inhibition of S-protein RBD (WT) binding to heparin using solution competition SPR by sulfated glycans (heparin, PPS, and MPS). S-protein RBD concentration was 500 nM. Error bars represent standard deviations from triplicate tests. (**A**,**B**) = heparin; (**C**,**D**) = PPS; (**E**,**F**) = MPS. Measured IC_50_ = 56 nM, 35 nM, and 9 nM for heparin, PPS, and MPS, respectively.

**Figure 5 pharmaceuticals-15-00258-f005:**
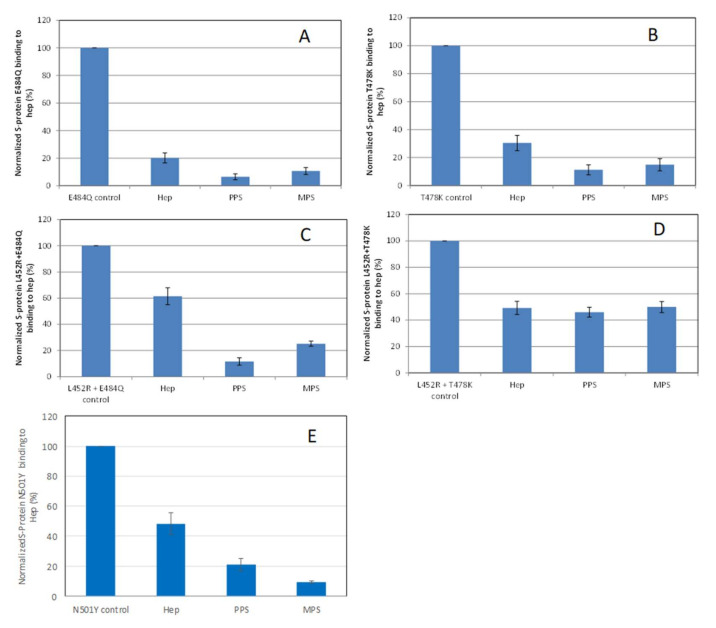
Bar graphs (based on triplicate experiments with standard deviation) of normalized S-protein RBD mutants: (**A**) E484Q; (**B**) T478K; (**C**) L452R+E484Q; (**D**) L452R+T478K; (**E**) N501Y binding to surface heparin by inhibiting with PPS and MPS or heparin at a concentration of 1000 nM in solution.

**Figure 6 pharmaceuticals-15-00258-f006:**
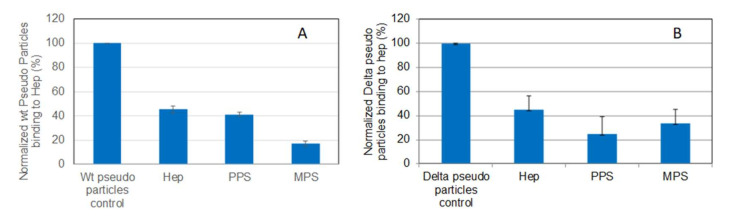
Bar graphs (based on triplicate experiments with standard deviation) of normalized pseudotyped viral particles binding to surface heparin under inhibition with RS or heparin. (**A**) Wild-type pseudo virus particles inhibited by heparin, PPS, and MPS; (**B**) Delta variant pseudotyped viral particles inhibited by heparin, PPS, and MPS.

**Figure 7 pharmaceuticals-15-00258-f007:**
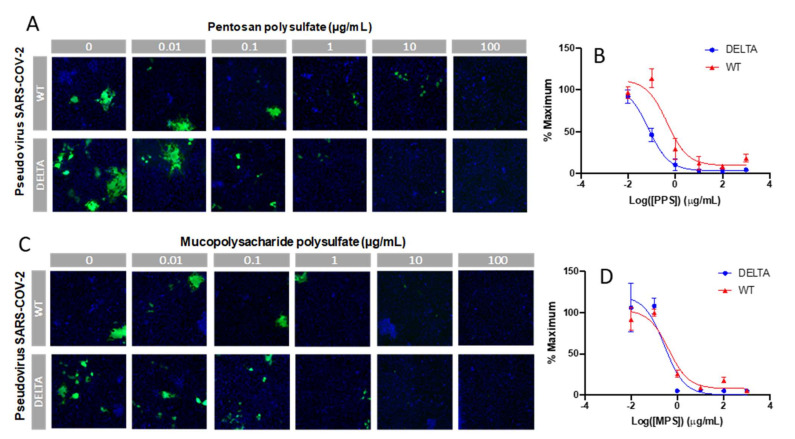
In vitro SARS-CoV-2 pseudotyped viral particle (WT and Delta variant) neutralization assay. (**A**,**C**) Representative fluorescence microscopy of different concentrations of PPS and MPS inhibition assay. (**B**,**D**) IC_50_ curves of PPS and MPS inhibiting SARS-CoV-2 pseudotyped virus (WT and Delta variant). Control experiments were performed using HA and dextran (MW: 12,000 Da), a charged, polyanionic glycosaminoglycan, and an uncharged polysaccharide, respectively. No antiviral activity was observed in either of these controls.

**Table 1 pharmaceuticals-15-00258-t001:** List of sulfated glycans or highly negative charged compounds.

Compound	Name	Manufacturers
#1	PPS: Pentosan polysulfate, MW: 6500 Da	Bene Pharma, Munich, Germany
#2	MPS mucopolysachharide polysulfate, MW: 14,500 Da	Luitpold Pharma, Munich, Germany
#3	Sulfated lactobionic acid, MW: 2400 Da	Luitpold Pharma, Munich, Germany
#4	Sulodexide, mixture of DS and Hep, MW: 7200 Da	Alfasigma, Milan, Italy
#5	Defibrotide 601, MW:~15,000 Da	Gentium, Milan, Italy
#6	Defibrotide 670, MW:15,000 Da	Gentium, Milan, Italy
#7	Defibrotide 648, MW: 16,200 Da	Gentium, Milan, Italy
#8	Defibrotide 0502, MW: 15,000 Da	Gentium, Milan, Italy
#9	4-t-butylcalix[6] arene-p-sulfonic acid: GL-2179. R = H	GeneLabs, Inc. Redwood, CA, USA
#10	4-t-butylcalix[6] arene-p-sulfonic acid: GL-2021, R = -CH_3_	GeneLabs, Inc. Redwood, CA, USA
#11	4-t-butylcalix[6] arene-p-sulfonic acid: GL-2029, R = acetyl	GeneLabs, Inc. Redwood, CA, USA
#12	4-t-butylcalix[8] arene-p-sulfonic acid: GL-288-Y-1	GeneLabs, Inc. Redwood, CA, USA
#13	4-t-butylcalix[8] arene-p-sulfonic acid: GL-522-Y-1, R = H	GeneLabs, Inc. Redwood, CA, USA
#14	4-t-butylcalix[8] arene-p-sulfonic acid: Calcium salt of GL-522Y-1	GeneLabs, Inc. Redwood, CA, USA

**Table 2 pharmaceuticals-15-00258-t002:** Summary of kinetic data of heparin and SARS-CoV-2 S-protein RBD (WT and VOC) interactions *.

Interaction	k_a_ *(1/MS)*	k_d_ *(1/S)*	*K_D_ (M)*
SARS-CoV-2 S-protein RBD wt	1427 (±26)	2.5 × 10^−4^ (±2.7 × 10^−6^)	1.8 × 10^−7^
SARS-CoV-2 S-protein RBD E484Q	5.2 × 10^4^ (±1.6 × 10^3^)	0.011 (±2.9 × 10^−4^)	2.0 × 10^−7^
SARS-CoV-2 S-protein RBD L452R+T478K	2.3 × 10^4^ (±100)	0.014 (±5.6 × 10^−4^)	5.9 × 10^−7^
SARS-CoV-2 S-protein RBD T478K	1.2 × 10^4^ (±180)	6.0 × 10^−3^ (±7.3 × 10^−5^)	4.9 × 10^−7^
SARS-CoV-2 S-protein RBD L452R	161 (±3.3)	1.3 × 10^−3^ (±5.4 × 10^−6^)	8.4 × 10^−6^
SARS-CoV-2 S-protein RBD N501Y	1.7 × 10^4^ (±81)	4.2 × 10^−4^ (±2.0 × 10^−6^)	2.5 × 10^−8^

* The data with (±) in parentheses represent standard deviations (SD) from the global fitting of five injections.

## Data Availability

Data is contained within the article.

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
