# Peer review of "Potential Anti-SARS-CoV-2 Activity of Pentosan Polysulfate and Mucopolysaccharide Polysulfate"

_pharmaceuticals, 2022, doi:10.3390/ph15020258_

Round 1
Reviewer 1 Report
This is an interesting study on the timely important issue.
In general, it can be accepted for publication.
2major criticisms are 1) Molecular docking analysis on the SARS CoV-2 RBD-ligand interaction can be simulated if they shorten the lengths of oligosaccharides but needs positive controls.
2) Cytotoxic effects on the host cells and also during interaction in vitro should be experimentally examined.
Author Response
Dear Reviewer,
Thanks for your positive and valuable comments on our manuscript. We have revised our manuscript carefully according to these comments. A point-by-point response to comments in the attached file.

Reviewer 2 Report
This manuscript describes the anti-SARS cov2 activity of sulfated glycans like pentosan polysulfate and mucopolysaccharide polysulfate using SPR analysis. In addition, authors describe in-vitro cell-based assay showing PPS and MPS exhibited strong anti-viral activities against SARS Cov-2. This is an organized manuscript that would be recommended for acceptance. Suggestions to improve this manuscript are listed below:
- The introduction section is missing some references
- Needs to have reference for statement on line 54,55 about the new anti-viral drugs from press release or publications
- Please add reference for glycosaminoglycans introduction about heparin and heparin sulfate. It was explained well in this article mentioned below. Please add this reference.
Citation: Lu, G: et al., Capillary Electrophoresis Separations of Glycans. Chemical Reviews. 2018, 118 (17), 7867-7885
- Please add reference for statement on line 84
- In Results and discussion section
- Line 119 discuss about Y501N mutant, but the figure discusses N501Y. Are these mutants both similar?
- The authors should include the Kd, Ka equations for the reader in the manuscript
- In section 2.2, figure 3 x-axis needs to have a label and guide the readers what does the number mean
- In section 2.3, can the authors mention which sugar moiety of glycan is binding to S-protein RBD for discussion on lines 156-158.
- In section 2.4, can you address the reader any reason for MPS & PPS being a stronger inhibitor than heparin?
- Grammatical errors need to be checked throughout the manuscript
- Please maintain the reference style format to be consistent for all the references.
- Some references do not have full journal name
- Some references do not have volume and page numbers
- If the reference is obtained in web, an access date is required
Author Response
Dear Reviewer,
Thanks for your positive and valuable comments on our manuscript. We have revised our manuscript carefully according to these comments. A point-by-point response to comments follows:
Response to Reviewer 2
This manuscript describes the anti-SARS cov2 activity of sulfated glycans like pentosan polysulfate and mucopolysaccharide polysulfate using SPR analysis. In addition, authors describe in-vitro cell-based assay showing PPS and MPS exhibited strong anti-viral activities against SARS Cov-2. This is an organized manuscript that would be recommended for acceptance. Suggestions to improve this manuscript are listed below:
- The introduction section is missing some references: Needs to have reference for statement on line 54,55 about the new anti-viral drugs from press release or publications
Response:
The reference has been added for the new anti-viral drugs.
- Please add reference for glycosaminoglycans introduction about heparin and heparin sulfate. It was explained well in this article mentioned below. Please add this reference.
Citation: Lu, G: et al., Capillary Electrophoresis Separations of Glycans. Chemical Reviews. 2018, 118 (17), 7867-7885. Please add reference for statement on line 84
Response: The reference (Lu, G: et al., Capillary Electrophoresis Separations of Glycans. Chemical Reviews. 2018, 118 (17), 7867-7885) has been cited in our revised manuscript.
- In Results and discussion section
Line 119 discuss about Y501N mutant, but the figure discusses N501Y. Are these mutants both similar?
Response: This was a typo, “Y501N” should be N501Y
- The authors should include the Kd, Ka equations for the reader in the manuscript
Response: The binding kinetic equations have been included in our revised manuscript.
- In section 2.2, figure 3 x-axis needs to have a label and guide the readers what does the number mean
Response: The compound information for the numbers on Figure 3 x-axis has been provided in the figure legend.
- In section 2.3, can the authors mention which sugar moiety of glycan is binding to S-protein RBD for discussion on lines 156-158.
Response: Based on a recent study (Bertini, et al. bioRxiv 2021.12.19.473359; doi: https://doi.org/10.1101/2021.12.19.473359), the 2,3-disulfated polyxylan oligosaccharide is the key sugar moiety of PPS binding to S-protein RBD. MPS has heparin-like molecular structure with a high level of sulfo groups allowing this glycan to interact with S-protein.
- In section 2.4, can you address the reader any reason for MPS & PPS being a stronger inhibitor than heparin?
Response: This could be due to the higher level of sulfation on MPS and PPS than heparin. The average heparin disaccharide contains ~2.7 sulfo groups while MPS disaccharide has > 4 sulfo groups and PPS disaccharide has > 3 sulfo groups.
- Grammatical errors need to be checked throughout the manuscript
Response: Grammatical errors have been carefully checked and fixed.
- Please maintain the reference style format to be consistent for all the references.
Some references do not have full journal name; Some references do not have volume and page numbers; If the reference is obtained in web, an access date is required.
Response: All the references are carefully checked and formatted.
Reviewer 3 Report
The authors claim to give an overview on the binding capacities of sulphated glycans towards ACE2 receptor. After a screening step, only two molecules (PPS and MPS) are tested in detail. Both show an interesting binding pattern. In addition, these molecules are already in medical use, so they are promising candidates for further application.
Major concern:
The experiments are well planned and well done. However, for the in vitro tests (Fig 7) I miss appropriate controls. At least one other charged molecule AND a non-charged polysaccharide should be included in order to make sure that the measured inhibition is not due to unspecific concentration effects.
Minors:
119 typo: Y501N should be N501Y I suppose.
Fig2: The black line (“summery”) should be thicker, now it is more or less invisible.
Fig4: Explain why for PPS are only 4 concentration points and for the others 5.
Fig4: A+B=heparin, C+D=PPS, E+F=MPS – give this information also in the figure, currently not easy to assign.
Author Response
Thanks for your positive and valuable comments on our manuscript. We have revised our manuscript carefully according to these comments. A point-by-point response to comments in the attached file.

Round 2
Reviewer 1 Report
The revision is appropriate but attenuated virions can be examined.
Generally, the present study is acceptable due to the instrumental inhibition of interaction between the receptor and ligands. However, there are commercially available viral models developed to check it.
Author Response
The revision is appropriate but attenuated virions can be examined.
Generally, the present study is acceptable due to the instrumental inhibition of interaction between the receptor and ligands. However, there are commercially available viral models developed to check it.
Response: Thanks for reviewer's positive and constructive comments.
The pseudovirus is a well-accepted model. There is no functional difference between a pseudovirus and an attenuated virus regarding the spike protein. Furthermore, our studies require dose response curves. Thus, the use of an expensive commercial attenuated virus, in the quantities required for our studies, would be cost prohibitive, unlike the pseudovirus that we generate ourselves. We also have no way to generate or propagate our own attenuated virus as this would require the use of either a BSL3 or a high level BSL2 lab to which we do not have access.
Reviewer 3 Report
Now its fine! Thank you!
Author Response
Thanks for the reviewer's positive comments.